# Immunotherapy Updates in Advanced Hepatocellular Carcinoma

**DOI:** 10.3390/cancers13092164

**Published:** 2021-04-30

**Authors:** Amisha Singh, Ryan J. Beechinor, Jasmine C. Huynh, Daneng Li, Farshid Dayyani, Jennifer B. Valerin, Andrew Hendifar, Jun Gong, May Cho

**Affiliations:** 1Internal Medicine, University of California, Davis, Sacramento, CA 95817, USA; drasingh@ucdavis.edu; 2UC Davis Comprehensive Cancer Center, Sacramento, CA 95817, USA; rbeechinor@ucdavis.edu; 3Hematology Oncology, University of California, Davis, Sacramento, CA 95817, USA; jachuynh@ucdavis.edu; 4Department of Medical Oncology, City of Hope Comprehensive Cancer Center and Beckman Research Institute, Duarte, CA 91010, USA; danli@coh.org; 5Hematology Oncology, University of California, Irvine, Irvine, CA 92868, USA; fdayyani@hs.uci.edu (F.D.); jvalerin@hs.uci.edu (J.B.V.); 6Hematology Oncology, Cedars-Sinai Medical Center, Los Angeles, CA 90048, USA; andrew.hendifar@cshs.org (A.H.); jun.gong@cshs.org (J.G.)

**Keywords:** hepatocellular carcinoma, immunotherapy, immune checkpoint inhibitors, PD-1, PD-L1, CTLA-4, chimeric antigen receptor

## Abstract

**Simple Summary:**

Advanced hepatocellular carcinoma (HCC) carries a grim prognosis, which has historically been compounded by a lack of available systemic therapies. Sorafenib monotherapy was the standard of care for front-line treatment of advanced HCC for many years, despite both poor tolerability and lack of durable responses. In the past few years, there have been several clinical trials evaluating the efficacy of immune checkpoint inhibitors for advanced HCC. Use of immune checkpoint inhibitors alone, and in combination with targeted therapies, has led to improved outcomes in both treatment-naïve and subsequent line treatment of advanced HCC. Here we review the role of immunotherapy in the treatment of HCC, describe the mechanistic basis for combination with targeted therapy, and summarize the recent published data as well as ongoing clinical trials for the use of immunotherapy in the treatment of advanced HCC.

**Abstract:**

Hepatocellular carcinoma (HCC) is the second most common cause of cancer death worldwide. HCC tumor development and treatment resistance are impacted by changes in the microenvironment of the hepatic immune system. Immunotherapy has the potential to improve response rates by overcoming immune tolerance mechanisms and strengthening anti-tumor activity in the tumor microenvironment. In this review, we characterize the impact of immunotherapy on outcomes of advanced HCC, as well as the active clinical trials evaluating novel combination immunotherapy strategies. In particular, we discuss the efficacy of atezolizumab and bevacizumab as demonstrated in the IMbrave150 study, which created a new standard of care for the front-line treatment of advanced HCC. However, there are multiple ongoing trials that may present additional front-line treatment options depending on their efficacy/toxicity results. Furthermore, the preliminary data on the application of chimeric antigen receptor (CAR-T) cell therapy for treatment of HCC suggests this may be a promising option for the future of advanced HCC treatment.

## 1. Introduction

Hepatocellular carcinoma (HCC) represents 90% of all primary liver tumors and is estimated to be the second most common cause of cancer deaths worldwide [1]. Advanced HCC carries a particularly poor prognosis, as median overall survival (OS) without treatment is approximately eight months [2]. Over the past five years, increased understanding of the pathogenesis and heterogeneity of these tumors has led to substantial progress in the development of systemic therapies for the treatment of HCC. Specifically, the use of immune checkpoint inhibitors alone, as well as in combination with targeted therapies, has proven to be an effective strategy for patients with advanced HCC. This review will discuss the theoretical context of immunotherapies and evaluate their practical significance in treating advanced HCC.

## 2. Role of Immunotherapy in HCC

The liver is a unique anatomical organ in terms of its role in promoting immune tolerance. It is fed by a dual blood supply from both the hepatic artery and portal vein, which facilitates gut pathogen exposure to Kupffer cells (macrophages), natural killer cells, and innate T cells in the hepatic sinusoids [3]. The constant exposure to these foreign antigens requires immune tolerance mechanisms mediated by regulatory T cells (Treg) and immunosuppressive cytokines to limit excessive immune activity to harmless antigens [4]. The hepatic immune system facilitates an immunosuppressive environment, which can promote the growth and prevent the immune capture of malignant hepatocytes, thereby making hepatic tumors a potential target for immunotherapy.

The pathogenesis of HCC is rooted in known inflammatory risk factors such as toxins, non-alcoholic hepatic steatosis, and viruses such as hepatitis A and B. HCC often arises on a background of cirrhosis related to these insults, as maladaptive interactions between angiogenic cells, fibroblastic cells, and immune cells promote pathologic tumor growth [5]. Imbalances between immune-suppressive and immune-activating cells play an important role in the development of HCC, and these immunosuppressive changes in the tissue microenvironment may also have prognostic implications. In vitro assays have shown that increased Treg expression correlates with an increasing stage of HCC tumors [6]. Studies have also shown that increased Treg expression correlates with poor prognosis and propensity for metastatic disease [7,8,9,10]. Tregs diminish immune activity by impairing effector T cell (CD8+) infiltration and by reducing granzyme and perforin activity [6]. Tregs, in tandem with myeloid-derived suppressor cells (MDSC), also suppress antiviral immune responses by upregulating the expression of checkpoint inhibitors such as programmed cell death protein 1 (PD-1) and programmed cell death protein ligand-1 (PD-L1) [11,12,13,14,15,16]. PD-L1 upregulation subsequently diminishes cytokine production, promotes Treg differentiation, and blunts cytotoxic responses from effector T cells [17]. There is significant literature to support that PD-L1 expression is associated with higher tumor stage, increased tumor recurrence risk, and worse overall prognosis [10,11,16,18]. This is further supported by in vitro studies that show positive responses to PD-L1 blockade on reducing viral load, preventing tumor-derived immunosuppression, and slowing tumor progression [12].

Based on the pre-clinical data suggesting the potential role of immunotherapy in the treatment of HCC, a multitude of clinical trials have been performed evaluating its use in patients with advanced HCC. A summary of the pivotal trials of immunotherapy for the treatment of advanced hepatocellular carcinoma is shown in Table 1. Collectively, these data have transformed the treatment of HCC from the previous standard of single agent targeted oral therapy with either sorafenib or lenvatinib, to a new standard of care using immunotherapy combinations. Below we review key clinical trial data and highlight important ongoing research that will likely impact advanced HCC management in the future.

## 3. Immune Checkpoint Inhibitor Monotherapy 

### 3.1. Nivolumab

Nivolumab is a fully humanized immunoglobulin G4 (IgG4) antibody against the PD-1 receptor. Through binding to PD-1, nivolumab prevents tumor cells from neutralizing T cell responses, thereby enhancing host T cell proliferation, increasing cytokine production, and leading to anti-tumor immune response [27]. The initial efficacy of nivolumab for the treatment of HCC was demonstrated in the CHECKMATE-040 trial, a multicenter, open-label phase I/II study of both sorafenib-naïve and sorafenib-treated patients with advanced HCC [19]. The trial was strict in its inclusion criteria, as only intermediate or advanced HCC patients with Child-Turcotte-Pugh class of A or better were included. Nivolumab was given intravenously (IV) every 2 weeks, and the study utilized a standard 3+3 design to determine the maximum tolerated dose ranging from 0.1–10 mg/kg, with the dose-expansion phase proceeding with 3 mg/kg IV. The objective response rate (ORR) was 20% (95% CI: 15–26) in the dose-expansion phase, and 15% (95% CI: 6–28) in the dose-escalation phase [19]. Based on these results, in 2017, nivolumab was FDA approved for treatment of advanced HCC after sorafenib failure.

Subsequently in 2019, the CHECKMATE-459 trial compared nivolumab to sorafenib as a first line therapy for advanced HCC [20]. This phase III study enrolled 743 systemic-therapy-naïve patients with advanced HCC, randomized to receive either nivolumab 240 mg IV every 2 weeks or oral sorafenib 400 mg by mouth twice a day. There was no statistically significant difference in the primary endpoint of median OS, which was 16.4 months with nivolumab versus 14.7 months with sorafenib (HR 0.85, 95% CI: 0.72–1.02; *p* = 0.0752). However, nivolumab demonstrated a better safety profile as adverse events ≥ grade 3 were reported in 81 patients (22%) receiving nivolumab versus 179 patients (49%) receiving sorafenib. In the nivolumab arm, patients with PD-L1 expression ≥1% demonstrated a higher ORR (28.2% vs. 12.2%) compared to those without, but this was not associated with PFS or OS benefit. However, comparing all enrolled patients with PD-L1 expression ≥1%, median OS was greater in patients receiving nivolumab compared to sorafenib (16.1 months vs. 8.6 months) [20]. This supports the theoretical notion that PD-L1 status may influence outcomes with anti-PD1 therapies such as nivolumab. Based on data from CHECKMATE-459, nivolumab monotherapy is not considered to be superior to sorafenib in the front-line treatment of advanced HCC. Therefore, its primary utility as a first line agent is in patients who are ineligible or intolerant to tyrosine kinase inhibitors or anti-VEGF treatments.

### 3.2. Pembrolizumab

Pembrolizumab is a fully humanized IgG4 kappa monoclonal antibody against PD-1 with a similar mechanism of action as nivolumab [28]. Pembrolizumab was granted accelerated approval by the FDA for advanced HCC as second-line treatment after sorafenib based on the KEYNOTE-224 trial [21]. This open-label, single-arm phase II trial enrolled 104 patients with advanced HCC with intolerance or progression with sorafenib, Child-Pugh A disease, and ECOG 0–1. Participants were treated with pembrolizumab 200 mg IV every three weeks until toxicity or progression. The primary endpoint of ORR was 17% (95% CI: 11–26). Median OS was 12.9 months (95% CI: 9.7–15.5), median PFS was 4.8 months (95% CI: 3.4–7.2), and grade ≥3 adverse effects were reported in 26% of patients [21].

Pembrolizumab was further evaluated for treatment of advanced HCC in the KEYNOTE-240 trial. This was a phase III, randomized, double-blind trial comparing pembrolizumab 200 mg IV every three weeks (*n* = 279) to placebo (*n* = 134) in 413 patients with advanced HCC previously treated with sorafenib [22]. As expected, patients receiving pembrolizumab demonstrated a higher ORR of 16.9% (95% CI: 12.7–21.8%) compared to placebo 2.2% (95% CI: 0.5–6.4%). Surprisingly, the coprimary endpoint of OS and PFS failed to reach the prespecified one-sided level of statistical significance (*p* = 0.0174 and *p* = 0.002, respectively) after a median follow-up of 13 months. The median OS was 13.9 months (95% CI: 11.6–16.0) in the pembrolizumab arm versus 10.6 months (95% CI: 8.3–13.5) in the placebo arm (HR 0.781; 95% CI: 0.611–0.998; *p* = 0.0238). PFS was similar, 3 months (95% CI: 2.8–4.1) for pembrolizumab versus 2.8 months (95% CI: 2,5–4.1) with placebo (HR: 0.775; 95% CI: 0.609–0.987; *p* = 0.0186). Treatment-related adverse events grade ≥3 occurred in 18.6% of patients in the pembrolizumab arm and 7.5% of patients taking placebo, demonstrating similar tolerability as with KEYNOTE-224 [21,22]. Given that the design of this study was a non-active comparator arm, it is surprising that OS/PFS was not improved with pembrolizumab compared to placebo. However, it is important to note that prior to study enrollment, there were no FDA approved medications for the treatment of HCC after progression on sorafenib. During the conduct of the trial, both regorafenib and nivolumab were approved as second line agents in advanced HCC after sorafenib, and the use of either or both of these agents at progression may have impacted these results.

### 3.3. Tremelimumab

Tremelimumab is a fully humanized IgG2 monoclonal antibody against cytotoxic T lymphocyte-associated antigen (CTLA4) [29]. CTLA-4 is an extracellular receptor expressed on T cells and is a CD28 homolog. In this fashion, CTLA-4 binds to B7 ligands expressed on antigen presenting cells (APC), and does so with a higher affinity than CD28. However, unlike CD28, which produces a costimulatory effect on T cells, CTLA-4 leads to T cell anergy, as it prevents the CD28/B7 costimulation required for T cell activation and proliferation. Therefore, CTLA-4 inhibitors facilitate immune-mediated anti-tumor response by antagonizing CTLA-4/B7 interactions, enabling CD28/B7 costimulatory interactions, and subsequently leading to direct activation and expansion of effector T cells which can target cancer antigens [28,29,30].

Tremelimumab was initially evaluated for use in advanced HCC in a 21-patient cohort of HCV-associated advanced HCC [31]. Patients were permitted to have previous treatment after a washout of at least 4 weeks, and 5/21 (23.8%) had previously received sorafenib. Patients were treated with tremelimumab at a dose of 15 mg/kg IV every 90 days until tumor progression or medication intolerance. The study noted a median time to progression of 6.5 months (95% CI: 3.95–9.14) with a median OS of 8.2 months (95% CI: 4.6–21.4). Importantly, tremelimumab demonstrated an acceptable toxicity profile in these patients with impaired liver function as well as a potential antiviral effect against HCV, and thus results help support future studies of CTLA-4 inhibitors in advanced HCC [31]. A recent phase II study (Study 22) that contained a tremelimumab monotherapy arm was presented at the 2020 American Society of Clinical Oncology (ASCO, Alexandria, VA, USA) Annual Meeting, and details of this trial are discussed later [32,33]. Additionally, the activity of tremelimumab when combined with radiofrequency thermal ablation (RFA) or transarterial chemoembolization (TACE) has also been evaluated for patients with advanced HCC [34]. Patients in this study were heavily pre-treated, as 21/28 (75%) had previously received sorafenib. Tremelimumab was given at two doses (3.5 or 19 mg/kg) IV every 4 weeks for 6 doses, then every 90 days until off treatment. Authors reported a median time to progression of 7.4 months (95% CI: 4.7–19.4) and median OS of 12.3 months (95% CI: 9.3–15.4) [34]. Based on this data, the combination of immunotherapy with direct disease interventions such as TACE or RFA represents a promising strategy for HCC, as these procedures expose tumor antigens, which may prime APCs for a more robust immune-mediated T cell response.

### 3.4. Durvalumab

Durvalumab is a humanized immunoglobulin G1k monoclonal antibody against the PD-L1 ligand [35]. Its mechanism is similar to that of pembrolizumab and nivolumab; however, durvalumab binds to the PD-L1 ligand, which is upregulated on cancer cells, rather than the PD-1 receptor on activated T cells. Inhibition of the PD-L1 ligand prevents it from downregulating T cells through its interaction with the PD-1 receptor, and thus leads to immune-mediated tumor cell recognition and killing [28,35]. The efficacy of durvalumab as monotherapy for patients with several tumor types, including advanced HCC, was investigated in a phase I/II, multicenter, open label study [36]. Patients enrolled (36/40) were heavily pre-treated with sorafenib. Patients were administered durvalumab 10 mg/kg IV every 2 weeks for 12 months or until complete response. Preliminary results, available in an abstract, demonstrated a median OS of 13.2 months (95% CI: 6.3–21.1) and a response rate of 4/39 (10.3%). Durvalumab was well-tolerated, as no patients discontinued therapy due to treatment associated adverse effects. This data demonstrated the potential efficacy of durvalumab in the treatment of advanced HCC and led to its ongoing study in combination with other immunotherapy discussed later.

## 4. Combination Strategies with Immune Checkpoint Inhibitors

### 4.1. Atezolizumab and Bevacizumab 

Among the ongoing clinical trials for advanced HCC, the combination of vascular endothelial growth factor (VEGF) inhibitors with immune checkpoint inhibitors may hold the greatest promise. HCC tumors’ carcinomas are heavily vascularized with rich arterial flows, driven in part by increased VEGF expression and angiogenesis [37]. In addition to its angiogenic effects, VEGF increases the recruitment of regulatory T cells, increases MDSC, and decreases the infiltration of effector T cells in tumors by inducing Fas ligand expression on tumor endothelium [38,39]. This effectively creates an inactive and walled off tumor microenvironment, and thus may limit the efficacy of immunotherapy alone. Furthermore, immune checkpoint inhibitors have been shown to promote normalization of vasculature. [40] Therefore, combination strategies targeting both VEGF and immune checkpoint molecules may be necessary to prevent MDSC- and Treg-mediated immunosuppression, strengthen effector T cell response, and ensure antiangiogenic effects in order to achieve optimal treatment outcomes [41].

Atezolizumab is a humanized IgG1 kappa immunoglobulin monoclonal antibody against PD-L1, and has an identical mechanism of action as durvalumab, described previously [28,42]. Atezolizumab was originally FDA approved in 2016 for metastatic non-small cell lung carcinoma, and it has subsequently been shown to be effective against many tumor types, including advanced urothelial carcinomas, triple negative breast cancer, metastatic melanoma, and HCC [43].

Bevacizumab is a humanized monoclonal IgG1 antibody that blocks VEGF and its downstream angiogenic and immunosuppressive effects. This blockade allows dendritic cell maturation and reduced regulatory T cell activity in the tumor microenvironment [44]. Bevacizumab has been used in combination with cytotoxic chemotherapy for a variety of solid malignancies and has only recently been investigated in combination with immunotherapy [45]. As both atezolizumab and bevacizumab target immunosuppressive regulatory T cell activity through different mechanisms, the combination of atezolizumab and bevacizumab is theorized to synergistically increase cytotoxic T cell attacks against tumor cells [41].

The combination of atezolizumab and bevacizumab was first investigated for the treatment of advanced HCC in a randomized phase Ib study (GO30140) comparing atezolizumab monotherapy to atezolizumab plus bevacizumab in 119 treatment-naïve patients [46,47]. The study enrolled 223 patients who were randomized either to atezolizumab 1200 mg IV every 3 weeks, with or without the addition of bevacizumab 15 mg/kg IV every 3 weeks (Arm F), or to a single-arm study of the combination of atezolizumab plus bevacizumab with the same dosing (Arm A). In Arm F, the median PFS was 5.6 months (95% CI: 3.6–7.4) in the patients treated with atezolizumab plus bevacizumab, compared to 3.4 months (95% CI: 1.9–5.2) in the atezolizumab monotherapy arm, which was statistically significant (HR 0.55, *p* = 0.0108). Grade ≥3 adverse effects were greater in the combination arm (20%) compared to atezolizumab monotherapy (5%). In Arm A, patients treated with atezolizumab plus bevacizumab had an ORR of 37/104 (36%). Adverse effects were notably higher in this single arm, as 39% of subjects had grade ≥3 adverse effects. These data suggested that this combination was a potential front-line treatment option for advanced HCC [46,47]. Subsequent evaluation of this combination was performed in the IMbrave150 trial, which was a Phase III open-label trial that enrolled 501 systemic-therapy-naïve patients [23,24]. Notably, unlike the KEYNOTE-240 trial, this trial allowed the inclusion of patients with main portal vein invasion. Patients were randomized to receive either atezolizumab 1200 mg IV plus bevacizumab 15 mg/kg IV every 3 weeks (*n* = 336) or sorafenib 400 mg twice a day (*n* = 165) until progression or intolerable toxicity. After a median follow-up period of 15.6 months, median OS was 19.2 months with combination therapy versus 13.4 months with sorafenib (HR, 0.66, 95% CI: 0.52–0.85, *p* = 0.0009). At 18 months, the survival rate was 52% in the combination cohort versus 40% with sorafenib. ORR was also higher with atezolizumab plus bevacizumab than with sorafenib (29.8% vs. 11.3%). The side effect profile was comparable between these regimens [23,24].

The results of the IMbrave150 trial set a new standard of care for the first-line treatment of advanced HCC, and there are several important implications of this trial that merit further discussion. First, atezolizumab plus bevacizumab is now the only therapy to demonstrate improved OS and PFS for the first-line treatment of advanced HCC compared to sorafenib. Importantly, the median OS of 13.4 months seen in the sorafenib arm of the IMbrave150 study is numerically greater than previous reports of front-line sorafenib use in both the SHARP study (10.7 months) and REFLECT study (12.3 months). This increase in OS occurred despite similar PFS seen in the sorafenib arms of the IMbrave150 study (4.3 months) when compared to that of the sorafenib arms in SHARP (5.5 months) and REFLECT (3.7 months) [2,23,48]. Therefore, the improved OS in the sorafenib arm of the IMbrave150 study occurred with a similar duration of sorafenib treatment, and this benefit likely represents the additional availability of second-line immunotherapy and TKIs in advanced HCC. Although the median OS with atezolizumab and bevacizumab was not reached in the original IMbrave150 publication, long term follow-up demonstrated this combination led to a 3.6-month advancement in median OS compared to sorafenib [24]. Secondly, this combination was well-tolerated, as the majority of adverse events were either infusion reactions or hypertension, and grade ≥3 bleeding rates as well as immune-mediated adverse events were rare (<10%). One reason for the low rate of bleeding may be that patients were excluded if they had untreated varices at baseline, and it will therefore be important in clinical practice to perform baseline upper endoscopies to mitigate this risk, as this was a requirement of the trial protocol [23,24]. To this point, it is also important to highlight the contraindications to this regimen. which include patients with autoimmune diseases, those with coinfection with hepatitis B or C, or those at high risk of bleeding. A third consideration is that only 15% of patients in IMbrave150 had Barcelona clinic liver stage B disease, and TACE is traditionally a consideration in this patient population. These patients have the potential for tumor downstaging to enable them to be candidates for surgery and/or transplantation, and thus their candidacy for this combination immunotherapy must be considered on a case-by-case basis. Lastly, the use of this immunotherapy combination up front will complicate the HCC treatment algorithm, as there is currently no available data to guide the treatment for patients who progress on this combination. For this reason, it is critically important that these patients be prioritized for clinical trials, as it is unknown whether therapies such as sorafenib or lenvatinib have efficacy after progression on an immunotherapy combination.

### 4.2. Tremelimumab and Durvalumab

The combination of CTLA-4 and PD-L1 inhibitors is believed to have the synergistic ability to overcome immune checkpoint resistance and induce anti-tumor activity [49]. A phase II multi-arm trial (Study 22) of durvalumab plus tremelimumab in 433 advanced HCC patients with intolerance or progression on sorafenib has shown promising results [32,33]. Patients in this study were treated with one of four possible regimens: Arm 1—tremelimumab 300 mg IV ×1 plus durvalumab 1500 mg IV every 4 weeks; Arm 2—durvalumab 1500 mg IV every 4 weeks; Arm 3—tremelimumab 750 mg IV every 4 weeks ×7 doses then every 12 weeks thereafter; or Arm 4—tremelimumab 75 mg IV ×4 plus durvalumab 1500 mg IV every 4 weeks. Reported median OS between the arms was as follows: 18.7 months, 13.6 months, 15.1 months, and 11.3 months for arms 1–4, respectively. Additionally, the reported ORR were 24.0%, 10.6%, 7.2%, and 9.5%, respectively. Toxicity data for these combinations is incomplete, but serious adverse events including death ranged from 10.9% up to 24.9% among the four arms [32,33]. Driven by the success of the phase II study, the ongoing phase III HIMALAYA trial (NCT03298451) with durvalumab and tremelimumab will test this combination as a first-line treatment for advanced HCC [50]. HIMALAYA is the first phase III study evaluating combined immune checkpoint inhibition in the first line treatment of advanced HCC. Patients will be randomized to one of three treatment arms: durvalumab monotherapy, durvalumab and tremelimumab combination therapy, or standard of care sorafenib. It is being conducted in over 186 centers across 16 countries, and the primary end point is OS [50]. If results demonstrate improved outcomes compared to sorafenib, it could provide another front-line treatment for advanced HCC and notably provide an alternative without antiangiogenic effects.

### 4.3. Nivolumab and Ipilimumab

Another dual immune checkpoint inhibitor combination, that of nivolumab and ipilimumab, was recently granted FDA approval for second-line therapy in sorafenib-refractory advanced HCC. Ipilimumab is a CTLA-4 monoclonal antibody with an identical mechanism of action as tremelimumab [18]. The combination of nivolumab and ipilimumab was studied in an expansion arm of the CHECKMATE-040 phase I/II trial, which compared outcomes among three dose combinations: Arm A was treated with nivolumab 1 mg/kg IV and ipilimumab 3 mg/kg IV every 3 weeks (*n* = 50); Arm B with nivolumab 3 mg/kg IV and ipilimumab 1 mg/kg IV every 3 weeks followed by nivolumab 240 mg IV every 2 weeks (*n* = 49); and Arm C with nivolumab 3 mg/kg IV every 2 weeks and ipilimumab 1 mg/kg IV every 6 weeks (*n* = 49) [51]. Therapy was generally well tolerated, and serious adverse events were rare. Arm A showed the most impressive outcomes with a 12-month OS rate of 61% (95% CI: 46–73%) and a 24-month OS rate of 48% (95% CI: 34–61%). Long-term results from a minimum 44-month follow-up period for Arm A showed an impressive median OS of up to 22 months and ORR of 32%. The objective response rates were similar across subgroups including hepatitis infection status, presence of vascular invasion and/or extrahepatic spread, and prior sorafenib treatment [51]. Interestingly, PD-L1 expression ≥1% was associated with an increased median OS (28.1 months vs. 16.6 months), whereas inflammatory biomarkers were not associated with differences in OS [52]. Similar to other combinations, results in this sorafenib-treated population have led to development of trials evaluating this combination in treatment-naïve patients. The open label phase III CHECKMATE 9DW trial comparing nivolumab plus ipilimumab versus sorafenib or lenvatinib as first-line treatment for advanced HCC is currently enrolling [53]. Depending on the results of the CHECKMATE 9DW and HIMALAYA trials, dual immune checkpoint inhibitor combinations could become another front-line treatment option for advanced HCC.

### 4.4. Tyrosine Kinase Inhibitor and PD-L1 Inhibitors

Lenvatinib is a small molecule tyrosine kinase inhibitor that opposes tumor angiogenesis by inhibiting VEGFR1-3, fibroblast growth factor receptor (FGFR1-4), and platelet-derived growth factor receptor α (PDGFRα) [54]. In 2018, Lenvatinib was granted FDA approval for first-line treatment of advanced HCC based on its efficacy demonstrated in the phase III REFLECT trial, where it was found to be non-inferior to sorafenib [48]. Given its multiple targets and efficacy as monotherapy, lenvatinib is currently being investigated in combination with immunotherapy for the front-line treatment of advanced HCC. The KEYNOTE-524 trial was an open-label, single arm, phase Ib study which evaluated the addition of pembrolizumab to lenvatinib in treatment-naïve patients with unresectable HCC [25,26]. The first 6 patients were evaluated in a dose-limited toxicity phase, and the remaining 100 patients were treated with an expansion phase of lenvatinib 8 mg (if <60 kg) or 12 mg (if >60 kg) by mouth daily plus pembrolizumab 200 mg IV every 3 weeks. Results show a median OS of 22 months (95% CI: 20.4-NE), median PFS of 8.6 months (95% CI: 7.1–9.7), and ORR of 36% (95% CI: 26.6–46.2). This combination is associated with a high degree of toxicity, as grade 3 adverse effects were reported in 67/100 (67%) of patients, including 3 deaths related to study treatment (all respiratory failure) [25,26]. Spurred by these results, the LEAP-002 trial is a multicenter phase III study that plans to enroll 750 patients assigned to receive either lenvatinib 8 or 12 mg by mouth daily with pembrolizumab 200 mg IV every 3 weeks or lenvatinib plus placebo. The study will look at primary outcomes of PFS and OS and seek support for the regulatory approval of this combination [55,56].

With their success in hematological malignancies, chimeric antigen receptor-modified T (CAR-T) cell therapies are now being employed against solid malignancies including HCC [57,58]. CAR-T cells are composed of an antigen recognition domain that guides cell affinity and immunogenicity, a hinge–spacer region that allows flexibility, a transmembrane domain that assists in cytokine release, and an intracellular domain that enables T cell activation and subsequent cytokine release. Newer generation CAR-T cells also consist of costimulatory molecules to enhance activation and cytokine release [59,60].

Glypican-3 (GPC-3) has been a popular tumor associated antigen (TAA) for CAR-T therapy in HCC [61]. Although GPC-3 expression is historically associated with a worse prognosis in HCC, its overexpression on tumor cells can be leveraged to increase the effectiveness of CAR-T therapy. Anti-GPC-3 CAR-T cells have demonstrated powerful cytotoxic responses in HCC cell lines and in vivo experiments. Clinical trials using anti-GPC-3 CAR-T cells are currently ongoing [61,62,63].

Alpha-fetoprotein (AFP) is another glycoprotein that is overexpressed by HCC tumor cells and is well-known as a tumor biomarker in HCC. Unlike GPC-3, which is a cell surface protein, AFP is processed intracellularly and presented on the cell surface by Class I MHC complexes [64]. To overcome this barrier, Liu et al. generated AFP CAR-T cells that specifically bind the AFP158–166 peptide complexed with human leukocyte antigen (HLA)-A*02:01. In vivo studies with xenograft models of human HLA-A*02:01+/AFP+ tumors showed significant regression with AFP CAR-T cells [65]. Currently, there are trials being developed to optimize the AFP T cell construct for treatment of advanced HCC [66].

Potential barriers to CAR-T efficacy in HCC include tumor neovascularization and tumor epithelialization, which may prevent adequate CAR-T cell infiltration. Resistance may also develop from proliferation of tumor cells that do not express the CAR-T target antigens. However, fourth generation CAR-T cells are being designed to enhance IL-12 production. This would significantly boost the immunogenicity and ability of CAR-T cells to target nearby antigen-negative cells [67,68]. Another strategy to overcome resistance is to create bispecific antibodies that can neutralize two antigens at a time or bring together a duo of T cells to enhance their immunogenicity. A phase I trial for ERY974, an anti-glypican 3/CD3 bispecific antibody, is currently recruiting for solid tumors including HCC [69]. Furthermore, as described above, the introduction of AFP CAR-T cells has paved the way for generating CAR-T cells that target intracellular antigens in solid tumors.

## 5. Management of Immune Related Adverse Events (irAEs)

Immune-related adverse events (irAEs) can affect a multitude of organ systems, have a unique delay in onset of toxicity, and present a particular challenge in patients with HCC due to the incidence of viral hepatitis and liver cirrhosis in this patient population [70,71]. While irAEs are generally reversible and dose-dependent, irAEs can be life threatening if not properly managed, as larger meta-analyses have reported the incidence of fatal irAEs to be approximately 0.64% (42 fatal irAEs in 6528 patients) [72] A comprehensive discussion of the management of irAEs is beyond the scope of our review, as there are several recent guidelines addressing this topic [73,74,75]. The incidence of irAEs in patients with HCC is no different than that of other cancer patients treated with immunotherapy, and therefore these guidelines are highly relevant [70].

However, it is worth mentioning the toxicity profile of the combination of bevacizumab with atezolizumab from the IMbrave150 trial, given its potential role as the new standard of care. Common adverse effects of Grade 3 or more with atezolizumab and bevacizumab included hypertension (15.2%), aspartate aminotransferase increase (7%), alanine transferase increase (3.6%), proteinuria (3%), and platelet count decrease (3.3%). Notably, patients receiving atezolizumab and bevacizumab experienced slightly fewer Grade 5 events compared to the sorafenib group (4.6% vs. 5.8%). Grade 5 events, though rare, included esophageal or other gastrointestinal hemorrhage, possibly attributable to the anti-VEGF effects of this combination [23]. Despite these adverse effects, compared to the sorafenib arm, patients treated with atezolizumab and bevacizumab had a shorter median duration of therapy (7.4 months vs. 8 months), achieved a higher median dose intensity of immunotherapy (95% vs. 85%), and a lower incidence of treatment related deaths (4.6% vs. 5.8%) [23].

Although there are detailed guidelines for the management of these toxicities, in general, bevacizumab should be discontinued for thromboembolic events, gastrointestinal perforation, or Grade 3 or higher hemorrhagic events. Furthermore, bevacizumab doses should be skipped rather than dose-reduced if patients experience adverse reactions. Atezolizumab should be discontinued for Grade 3 or higher pneumonitis or hepatitis. Additionally, it should be discontinued for any recurrent Grade 3 adverse effects. For atezolizumab-related colitis, there is a role for corticosteroids with Grade 2 or 3 toxicities, though treatment should be discontinued for Grade 4 colitis [73,74,75].

## 6. Future Directions

There are several key ongoing trials for the use of immunotherapy in HCC shown in Table 2. These include optimizing the use of biomarkers to identify responders to immunotherapy, overcoming resistance, and the potential role of immunotherapy in adjuvant HCC, each of which are detailed below. 

### 6.1. Emerging Biomarkers for Monitoring Response to Immunotherapies in HCC

With an influx of clinical trials studying novel checkpoint inhibitor combinations, PD-L1 expression is being increasingly studied as a biomarker for monitoring response to immunotherapy. Although previous studies have correlated PD-L1 expression with poor prognosis in advanced HCC, data from CHECKMATE-040 and KEYNOTE-224 trials do not demonstrate any strong correlation between PD-L1 expression and ORR [21,51]. However, there is significant inter-institutional variability in the methods used to determine PD-L1 expression, which could confound data from multicenter clinical trials. Ideally, participating institutions should also set uniform sensitivity cutoffs for determining positive PD-L1 status. Furthermore, as PD-L1 expression is dynamic with disease progression, it would be important to sub-classify advanced HCC patients to allow comparison between patients with similar tumor burdens. To this point, there is currently no data delineating the optimum timing for performing immunohistochemical analysis on HCC tissue samples to determine PD-L1 expression. Additionally, the source of PD-L1 expression (stromal vs. tumoral vs. combined) is yet to be standardized. Overall, the variability in the immunohistochemical analyses of PD-L1 expression limits its practical utility for assessing response to immunotherapy [81].

As the landscape of HCC tumors is constantly shaped by competing immune-suppressive and immune-activating players, it is reasonable to monitor tumor-infiltrating lymphocyte density to gauge the expected responses to treatment. Indeed, high intratumoral densities of CD3+ and CD8+ T cells have been associated with a longer recurrence-free survival. A study by Kaseb et al. showed that, in patients treated with nivolumab and ipilimumab, clinical responses correlated with CD8+ infiltration [82]. However, as T cell infiltration is expected to increase with chronic viral infection, the presence of chronic hepatitis C infection could confound these results. More research is needed to determine whether this class of patients would derive greater benefit from checkpoint inhibitors.

Another way to measure a tumor’s immunogenic potential is to measure its tumor mutational burden. Tumor mutational burden (TMB) represents the number of somatic mutations present in a tumor genome, with a high TMB defined as ≥10 mutations per mega-base (mut/mb) [83,84]. While TMB is emerging in popularity as a biomarker in various cancers, the relatively low TMB in HCC (median TMB of 5 mut/mb) has impeded its applicability. A comprehensive genomic profiling study of 755 advanced HCC patients by Ang et al. found the median TMB for the entire cohort to be on the lower end with 4 mut/mb. The authors also found no significant correlation between TMB and responders, progressors, or stable disease; this underscores the limited value of TMB as a current biomarker in HCC [85].

As HCC tumors are heterogeneous with distinct tumor microenvironment phenotypes, it is perhaps beneficial to classify tumors into grouped genetic profiles as opposed to individual biomarker classes. For instance, Thorsson et al. conducted an immunogenomic analysis and classified tumors into six different phenotypes. Four genetic “clusters” were identified as most common: lymphocyte depleted, inflammatory, wound healing, and interferon-γ dominant. In this manner, understanding the clustering of genes that helps create specific immune escape mechanisms could help guide the most synergistic combination therapies [86]. Creating grouped genetic profiles could also allow researchers to identify subgroups of patients who could benefit from targeted combination therapies.

### 6.2. Overcoming Resistance to Immunotherapy

Novel combination regimens of immunotherapy with targeted therapy carry the promise of decreasing the rate of immune checkpoint inhibitor resistance. In addition to the published results at the time of this review, there are a multitude of ongoing clinical trials that will add to this literature in the near future (Table 2). Furthermore, it is possible that optimization of these combinations may enhance response rates in patients who have progressed on standard immune checkpoint inhibitor regimens. Several VEGF plus TKI plus immune checkpoint inhibitor combinations have shown efficacy in those previously treated with immune checkpoint inhibitors in other tumor types such as renal cell carcinoma, and these could be potentially adapted to HCC [87]. An example of this is the combination of lenvatinib plus pembrolizumab, which is being evaluated in metastatic renal cell carcinoma patients who progressed on immune checkpoint inhibitor treatment [88].

There are a variety of other immune targets that hold promise for future drug development in HCC and may help mitigate resistance to immune checkpoint inhibition. One example is transforming growth factor (TGF-β), which plays a role in modulating regulatory CD4+ T cell interaction with malignant hepatocytes. Pre-clinical data has shown that TGF-β1 inhibition can overcome primary resistance to immune checkpoint blockade in an animal model without toxicity concerns noted for pan TGF-β inhibitors [89]. A second is the lymphocyte activation gene-3 (LAG-3), which has multiple effects on T cell function and has been shown to be upregulated in CD8+ lymphocytes in patients with HCC [90]. Additionally, T cell immunoglobulin and mucin domain containing-3 (TIM-3) is a protein expressed on CD8+ lymphocytes. This may represent a druggable target to enhance immune-mediated tumor response, and there are currently ongoing trials evaluating this strategy [91,92]. Lastly, tumor-derived extracellular vesicles (EVs), which mediate cell-to-cell interactions, have been shown to express PD-L1 and may contribute towards T cell suppression [93]. The impact of EVs has been shown to be relevant in HCC, as EV-derived molecules such as ubiquitin-like with PHD and RING finger domain (UHFR1) induce NK cell dysfunction in HCC patients [94].

### 6.3. Adjuvant Immunotherapy in HCC

There are multiple ongoing clinical trials evaluating the adjuvant use of immunotherapy in patients with intermediate stage HCC. As mentioned previously, there is preliminary evidence that tremelimumab combined with either TACE or RFA may represent a new therapeutic strategy for patients with earlier stage disease [34]. This strategy is being investigated in the IMbrave150, trial which is a randomized, open label, phase III study comparing atezolizumab and bevacizumab versus active surveillance in HCC patients after curative resection or RFA [95]. Additionally, the EMERALD-1 and EMERALD-2 trials are both randomized, double-blind, placebo-controlled phase III studies comparing durvalumab with or without bevacizumab to placebo in intermediate stage HCC patients receiving TACE or RFA [96,97]. If these studies show benefit for the use of adjuvant immunotherapy in HCC, this would further complicate treatment pathways, as the ideal treatment after progression on immunotherapy remains unclear.

## 7. Conclusions

The clinical trial data described above demonstrates limited efficacy for checkpoint inhibitor monotherapy for advanced HCC. However, combination therapies with checkpoint inhibitors and anti-VEGF treatments have shown promise for improving the poor survival data for advanced HCC. As we are just beginning to understand the applicability of immunotherapies to HCC, several new immunotherapy combinations are yet to be studied.

## Figures and Tables

**Table 1 cancers-13-02164-t001:** Published pivotal trials of immunotherapy for the treatment of advanced hepatocellular carcinoma.

Trial Name	Phase	Setting	Target	Intervention	* Key Results
CHECKMATE-040 [19]	Phase I/II	Previous progression or intolerance to sorafenib	PD-1 inhibitor, CTLA-4 inhibitor	Nivolumab + ipilimumab (Arm A)	ORR: 32%, CR: 5% Median PFS: NA, Median OS: 22.8 months
CHECKMATE-459 [20]	Phase III	First-line therapy for treatment-naïve	PD-1 inhibitor	Nivolumab	ORR: 15%, CR: 4%, Median PFS: 3.6 months, Median OS: 16.4 months
KEYNOTE-224 [21]	Phase II	Previous progression or intolerance to sorafenib	PD-1 inhibitor	Pembrolizumab	ORR: 17%, CR: 1%, Median PFS: 4.9 months, Median OS: 12.9 months
KEYNOTE-240 [22]	Phase III	Previous progression or intolerance to sorafenib	PD-1 inhibitor	Pembrolizumab	ORR: 18.3%, CR: 2.2%, Median PFS: 3.0 months, Median OS: 13.9 months
IMbrave150 [23,24]	Phase III	First-line therapy for treatment-naïve, comparison to sorafenib	VEGF inhibitor, PD-L1 inhibitor	Atezolizumab + bevacizumab	ORR: 27.3%, CR: 5.5%, Median PFS: 6.8 months, Median OS: 19.2 months
KEYNOTE-524 [25,26]	Phase Ib	DLT then expansion group of first-line therapy for treatment-naïve	TKI, PD-1	Lenvatinib + pembrolizumab	ORR: 46.0%, CR: 5.0%, Median PFS: 9.3 months, Median OS: 22 months

Abbreviations: PD-1, programmed cell death protein 1; CTLA-4, cytotoxic T-lymphocyte-associated protein 4; VEGF, vascular endothelial growth factor; PD-L1, programmed death ligand 1; TKI, tyrosine kinase inhibitor; DLT, dose limiting toxicity; ORR, objective response rate; CR, complete response; PFS, progression free survival; OS, overall survival. * Results shown are those reported using RECIST 1.1.

**Table 2 cancers-13-02164-t002:** Ongoing pivotal trials of immunotherapy for the treatment advanced hepatocellular carcinoma.

Trial Name	Phase	Setting	Target	Intervention	Study Details
CHECKMATE-9DW [53]	Phase III	First-line therapy for treatment-naïve, comparison to sorafenib or lenvatinib	PD-1 inhibitor, CTLA-4 inhibitor	Nivolumab + ipilimumab	Enrollment: 650 participants Estimated Completion: September 2023
NCT03764293 [76]	Phase III	First-line therapy for treatment-naïve, comparison to sorafenib	TKI, PD-1 inhibitor	Camrelizumab (SHR-1210) + apatinib	Estimated Enrollment: 510 participants Estimated Completion: June 2022
COSMIC-312 [77]	Phase III	First-line therapy for treatment-naïve, comparison to sorafenib	TKI, PD-L1 inhibitor	Atezolizumab + cabozantinib	Estimated Enrollment: 740 participants Estimated Completion: December 2021
HIMALAYA [50,78]	Phase III	First-line therapy for treatment-naïve, comparison to sorafenib	PD-1 inhibitor, CTLA-4 inhibitor	Durvalumab + tremelimumab	Estimated Enrollment: 1504 participants Estimated Completion: April 2022
LEAP-002 [56]	Phase III	First-line therapy for treatment-naïve, comparison to lenvatinib	TKI, PD-1 inhibitor	Lenvatinib + pembrolizumab	Estimated Enrollment: 750 Estimated Completion: May 2022
ORIENT-32 [79]	Phase II/III	First-line therapy for treatment-naïve, comparison to sorafenib	PD-1 inhibitor, VEGF	Sintilimab + IBI308	Estimated Enrollment: 595 Estimated Completion: December 2022
RATIONALE-301 [80]	Phase III	First-line therapy for treatment-naïve, comparison to sorafenib	PD-1 inhibitor	Tislelizumab	Estimated Enrollment: 674 Estimated Completion: May 2022

Abbreviations: PD-1, programmed cell death protein 1; CTLA-4, cytotoxic T-lymphocyte-associated protein 4; TKI, tyrosine kinase inhibitor; PD-L1, programmed death ligand 1; CAR-T, chimeric antigen receptor-modified T cell.

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
