# Peer review of "Immunotherapy Updates in Advanced Hepatocellular Carcinoma"

_cancers, 2021, doi:10.3390/cancers13092164_

Round 1

Reviewer 1 Report

After taking a look at the revised manuscript, I recommend a favourable decision as my previous comments have been well addressed. 

Please proceed.

Reviewer 2 Report

I have no further comments.

This manuscript is a resubmission of an earlier submission. The following is a list of the peer review reports and author responses from that submission.

Round 1

Reviewer 1 Report

The manuscript by Singh et al is a well written and comprehensive review that summarises the immunotherapy updates in advanced HCC. The authors review the key monotherapy options and highlight the pivotal combination strategies in treating advanced HCC. The authors also provide their unique insight into the future directions of this topic. The review will be a valuable and timely update to medical oncologists, pathologists, and other researchers interested. I suggest the following points for the authors’ consideration to enhance the manuscript.

  • Despite the acceptable tolerability and safety profiles of some immunotherapy drugs, adverse effects related are an emerging issue in the clinic. Adding a subsection discussing this in Section 6 would make the review complete and more comprehensive, especially for the combination of atezolizumab and bevacizumab that has become the new first line for advanced HCC.
  • Extracellular vesicles are increasingly recognised as a contributor to immunotherapy resistance (PMID: 33545339) and presents novel targeting strategies. The authors are encouraged to take this aspect of resistance development into consideration.
  • A short transitional paragraph between Section 2 and Section 3 would make this part more coherent and cohesive.
  • In Line 217 – 220, would the anti-angiogenic effect be a potential goal of combination strategy as well?
  • In Line 317, please use one or two sentences to briefly introduce ipilimumab as it is first mentioned here.

Reviewer 2 Report

This review summarized immunotherapy and combination of immunotherapy with target therapy in treatment of advanced HCC. Generally it is well-written. The major issue is where is the target audience for this review. This review summarized details of many published studies and ungoing trials only and lack a specific focus. It is more like a collection of clinical data than a re-organized review. As many authors are speakers or advisors in multiple phama companies, this so-called "theoretical" review can be seen everywhere.